# Expression of Human Mutant Preproinsulins Induced Unfolded Protein Response, *Gadd45* Expression, JAK-STAT Activation, and Growth Inhibition in *Drosophila*

**DOI:** 10.3390/ijms222112038

**Published:** 2021-11-07

**Authors:** Tatsuki Yamazoe, Yasuyuki Nakahara, Hiroka Katsube, Yoshihiro H. Inoue

**Affiliations:** Department of Insect Biomedical Research, Kyoto Institute of Technology, Matsugasaki, Kyoto 606-0962, Sakyo, Japan; tatsuki.yamazoe@gmail.com (T.Y.); potte53@yahoo.co.jp (Y.N.); hii.luk1226@gmail.com (H.K.)

**Keywords:** ER stress, *Gadd45*, JNK, diabetes, NDM, *Drosophila*

## Abstract

Mutations in the insulin gene (INS) are frequently associated with human permanent neonatal diabetes mellitus. However, the mechanisms underlying the onset of this genetic disease is not sufficiently decoded. We induced expression of two types of human mutant INSs in *Drosophila* using its ectopic expression system and investigated the resultant responses in development. Expression of the wild-type preproinsulin in the insulin-producing cells (IPCs) throughout the larval stage led to a stimulation of the overall and wing growth. However, ectopic expression of human mutant preproinsulins, hINS^C96Y^ and hINS^LB15YB16delinsH^, neither of which secreted from the β-cells, could not stimulate the *Drosophila* growth. Furthermore, neither of the mutant polypeptides induced caspase activation leading to apoptosis. Instead, they induced expression of several markers indicating the activation of unfolded protein response, such as ER stress-dependent *Xbp1* mRNA splicing and ER chaperone induction. We newly found that the mutant polypeptides induced the expression of *Growth arrest and DNA-damage-inducible 45* (*Gadd45*) in imaginal disc cells. ER stress induced by hINS^C96Y^ also activated the JAK-STAT signaling, involved in inflammatory responses. Collectively, we speculate that the diabetes-like growth defects appeared as a consequence of the human mutant preproinsulin expression was involved in dysfunction of the IPCs, rather than apoptosis.

## 1. Introduction

Diabetes is a group of metabolic diseases wherein patients show a condition of elevated blood sugar levels called hyperglycemia. This disease is roughly classified into three principal types: type 1 diabetes, type 2 diabetes, and gestational diabetes [1]. Among them, type 1 diabetes is considered to be an autoimmune disease characterized by inflammatory responses that leads to the gradual destruction of pancreatic β-cells. This type of cell damage causes insulin deficiency, and results in deregulation of glucose metabolism [2]. Neonatal diabetes mellitus (NDM) is a rare disorder associated with defects in mass and function of pancreatic β-cells. NDM is not an autoimmune disease but a monogenic form of diabetes resulting from mutations in several genes encoding proteins crucial for the normal activity of the pancreatic β-cells. Mutations in the human insulin gene (*INS*) are a common cause of permanent neonatal diabetes mellitus (PNDM). When some of these mutant proinsulin genes are expressed in human cultured cells, folding and/or secretion of the mutant polypeptides are disrupted [3,4,5]. Exogenous insulin therapy is the only treatment for PNDM patients with the INS mutations. Hence, it is crucial to identify factors involved in the pathogenesis and understand the mechanisms by which the onset of PNDM using animal models for development of more effective therapeutic agents.

It has been considered that the onset of NDM is caused by the dysfunction or destruction of pancreatic β-cells [6,7,8,9]. It was recently suggested that various types of cellular stresses other than autoimmune responses, especially that of the endoplasmic reticulum (ER), are associated with the destruction or the malfunctioning of β-cells [10,11,12]. However, the mechanism by which ER stress triggers dysfunction and/or destruction of β-cells remains unidentified.

Excessive accumulation of unfolded or misfolded proteins, produced by exceeding the folding capacity of chaperones, causes a stress condition called ER stress, and activates the Unfolded Protein Response (UPR). Three principal branch cascades involved in the UPR were identified previously [13], and each branch is controlled by a different sensor protein, namely inositol-requiring enzyme 1 (IRE1), double-stranded RNA-activated protein kinase-like ER kinase (PERK), and activating transcription factor 6 (ATF6). These proteins sense the accumulation of unfolded proteins in ER, and stimulate the transcription of genes encoding ER chaperones, such as GRP78. Furthermore, cell death is induced in cases of continuous stress accumulation that failed to be managed by the UPR. ER stress-induced apoptosis is associated with many human diseases, including neurodegenerative diseases and diabetes [14]. Moreover, it was reported that ER stress-induced UPR triggers the induction of several stress-associated genes. Among them, *Gadd45* (Growth Arrest and DNA-damage-inducible 45) plays a crucial role in DNA repair, cell cycle arrest, and apoptosis. It is induced by various genotoxic stresses through the tumor suppressor protein, p53 in mammalian cells [15]. Recent studies presented evidence suggesting that one of the Gadd45 family proteins, Gadd45α, was induced by ER stress through the PERK-eIF2α pathway in mouse embryonic fibroblasts [16]. However, the relationship between UPR and this *Gadd45* gene in living organisms remains largely unexplored.

Tissues in which many secreted proteins are synthesized (like pancreas) are susceptible to the induction of ER stress [17]. Therefore, excessive ER stress or attenuation of the UPR in IPCs could cause functional inhibition of the cell functions, resulting in reduced secretion of insulin, eventually leading to onset of diabetes. It, however, is uncertain whether the ER stress and attenuation of the UPR promote the development of the disease. Extensive studies using animal models are thus required to investigate this causal relationship.

*Drosophila melanogaster* has emerged a powerful model organism for many human diseases for the past decade, such as metabolic diseases including diabetes [18,19]. Insulin, its receptor, and its downstream signaling pathway are highly conserved between *Drosophila* and mammals [18,20]. 8 types of insulin-like peptides (Dilp1–8) were identified in *Drosophila* [21,22,23,24]. Of these, all Dilps except Dilp8 bind to a unique insulin receptor, termed InR, and trigger the conserved signaling cascade. Further, three types of Dilps are found to be expressed in 7 insulin-producing neurons in a hemisphere of the brain. The *Drosophila* insulin-producing cells (IPCs) work as counterparts of pancreatic β-cells in mammals [25,26]. Additionally, the experimental techniques to identify the IPCs and to induce a specific gene expression in the cells are established in this organism [18]. Therefore, we selected this model organism to address the issue of whether ER stress in the insulin-producing cells contributed to the pathogenesis of diabetes. Thus, investigation of these conserved mechanisms can further improve our understanding of the connections between ER stress and IPC dysfunction, and the factors that lead to diabetes.

Previously, Katsube and colleagues demonstrated that the expression of a dominant-negative mutant of Hsc70-3 (Hsc70-3^DN^), a *Drosophila* ER chaperone, induces ER stress in *Drosophila* tissues [27]. Using this Hsc70-3^DN^-induced ER stress model, the authors demonstrated that ER-stress resulted in the apoptosis of IPCs leading to the onset of the diseases. However, further studies are required to decide whether apoptosis in the IPCs is involved in the pathogenesis of PNDM under physiological condition.

In this study, to understand the mechanisms underlying the onset of PNDM at the organism level, we induced the expression of two types of human mutant preproinsulin identified from PNDM patients, hINS^C96Y^ (*Akita* mutation identified from PNDM mice introduced into human preproinsulin) and hINS^LB15YB16delinsH^ (identified from human PNDM patients) in *Drosophila* tissues during development. Neither of the human mutant peptides could be secreted from cultured cells derived from pancreatic β-cells, when they were ectopically induced [3]. We investigated whether expression of the human mutant preproinsulins induced the UPR in the *Drosophila* tissues. Moreover, as the relationship between UPR and the Gadd45 had been unexplored in *Drosophila*, we first examined whether the *Gadd45* gene expression depends on p53 in the tissues, similar to that in mammalian cultured cells. Then, we investigated whether the *Gadd45* transcription was induced in tissues that have an accumulation of ER stress. Here, we demonstrated that the Gadd45-GFP reporter was an excellent tool that monitors ER-stress response. Furthermore, we investigated whether the human mutant preproinsulins induced activation of UPR, and the JAK-STAT signaling that potentially regulates inflammatory responses. The c-Jun NH_2_-terminal kinases (JNK), known as one of the stress-activated MAP kinases (SAPK), is also related to UPR. The results from our investigation collectively suggested the crucial role of Gadd45 or JAK-STAT pathway in tissue dysfunction or destruction in organisms as a result of ER stress.

## 2. Results

### 2.1. Ectopic Expression of Human Mutant Preproinsulins Induced the Expression of Two Types of ER Stress Marker in Drosophila Tissues

To clarify whether the expression of human mutant preproinsulin was responsible for causing the PNDM, we induced the expression of the mutant proteins hINS^C96Y^ and hINS^LB15YB16delinsH^ (Appendix A) in *Drosophila melanogaster*. We investigated whether the ectopic expression of these polypeptides resulted in tissue dysfunction and developmental abnormality, which are reminiscent of diabetes. To achieve this, first, we induced targeted expression of *Drosophila* insulin-like peptide 2 and human preproinsulins in Insulin Producing Cells (IPCs) in *Drosophila* adults using the Gal4/UAS system. We then investigated whether normal and mutant preproinsulins were successfully synthesized and secreted from the cells in *Drosophila* adults. Upon overexpression of Dilp2 (*Drosophila* insulin-like peptide 2) in IPCs using an IPC-specific Gal4 driver, *ilp2-Gal4*, a stronger anti-Dilp2 immunostaining signal in the cells was observed after 12 h starvation (Appendix A). Compared with the Dilp2 signal in the IPCs, the signal became weaker due to its release from the cells in adults re-fed for 2 h after the starvation (*n* > 7, *p* < 0.05, Student’s *t*-test) (Appendix A). Consistently, immunostaining signal with a specific antibody against the human polypeptide was detected in the IPCs of *Drosophila* adults starved in the same way, harboring the cell-specific expression of wild-type hINS (*ilp2* > *hINS^WT^*) (*n* > 11, *p* < 0.01, Student’s *t*-test) (Appendix A). Here also, the signal consequently became weaker after re-feeding the adults with the diet (Appendix A). These observations suggested that it was possible to express the human wild-type insulin in *Drosophila* IPCs, and its release could be controlled by modulation of the diet. Then, we examined whether a human mutant polypeptide hINS^C96Y^ having the Akita mutation identified in PNDM model mice [3]. This mutant polypeptide has a Cys to Tyr substitution at proinsulin position 96 (C96Y; encoding residue 7 of the insulin A-chain), which disrupts a disulfide bond connecting the insulin B and A-chains (Appendix A). It was accumulated in the IPCs and released from the cells depending on nutrient condition. Upon expressing the human mutant insulin, in the IPCs, we detected the immunostaining signal over background level in the cells from the starved adults (*ilp2* > *hINS^C96Y^*) (Appendix A). However, a significant reduction of the signal was not observed in the IPCs expressing the human mutant insulin after re-feeding the diet (*n* > 14, *p* = 0.17, Student’s *t*-test) (Appendix A). This observation corroborated the published result that the mutant insulin failed to be secreted from cultured cells derived from pancreatic β-cells [3].

Next, we investigated whether the ectopic expression of the human polypeptides induced ER stress in the imaginal disc cells using an ER stress reporter Xbp1*-GFP. We used wing imaginal discs because they were easier to observe GFP expression than IPCs. The ER stress triggers splicing of *Xbp1* pre-mRNA (*Xbp1**) encoding a truncated Xbp1 precursor to remove an extra exon harboring a stop codon. If the full-length Xbp1-GFP protein is synthesized from a mature *Xbp1* mRNA in response to the stress, GFP fluorescence can be seen. In the control wing discs (*Bx-Gal4/+*), GFP fluorescence was not detected due to the absence of ER stress-induced splicing (Figure 1A). Similarly, *Bx-Gal4*-dependent expression of human wild-type preproinsulin (hINS^WT^) did not induce Xbp1*-GFP expression in the discs, suggesting the lack of ER stress (Figure 1B). In contrast, intense GFP fluorescence was observed specifically in the wing pouch region of wing imaginal discs, in which the ectopic expression of the human mutant polypeptide, hINS^C96Y^ or hINS^LB15YB16delinsH^, both of which failed to be secreted from β-cells, was induced (Appendix A). These human polypeptides have different amino acid substitutions identified from NDM model mouse and patients [3]. In addition to the hINS^C96Y^ described before, another human mutant insulin, hINS^LB15YB16delinsH^ carrying an in-frame substitution, in which B-chain Leu^15^ and Tyr^16^ are replaced by a single His residue. Both mutant polypeptides failed to secrete from β-cells. These observations indicate the induction of the ER stress (in *Bx* > *hINS^C96Y^*, *Xbp1*-GFP* or *Bx* > *hINS^LB15YB16delinsH^*, *Xbp1*-GFP*) (Figure 1C,D). We consistently observed distinctive GFP fluorescence in the wing disc region expressing the mutant peptides, although hINS^LB15YB16delinsH^ produced less intense fluorescence than hINS^C96Y^. Thus, both human mutant preproinsulins hINS^C96Y^ and hINS^LB15YB16delinsH^ can induce the UPR in *Drosophila* tissues. Especially, the former polypeptide causes a stronger response.

Next, we questioned whether the expression of the mutant polypeptides in IPCs resulted in the pathogenesis of diabetes-like phenotype via ER stress accumulation in the cells. Previously studies reported that *Drosophila* ER chaperone(s) and Hsc70 family proteins recognized by anti-GRP78 antibody are up-regulated under these stress conditions [27,28]. We induced these mutant polypeptides in IPCs of *Drosophila* larvae using *ilp2-Gal4.* We performed immunostaining of whole larval brains from control larvae (*ilp2-Gal4/+*) (Figure 1E), or brains expressing normal human preproinsulin or the mutant polypeptides (Figure 1F–H). The anti-GRP78 immunostaining signal was not observed in control IPCs without expression of the human polypeptides (*ilp2-Gal4/+*) (Figure 1E”). Similarly, in the larval IPCs expressing hINS^WT^ (*ilp2* > *hINS^WT^*), the immunostaining signal was similar to the background levels (Figure 1F”). In contrast, a relatively higher signal was seen in IPCs expressing hINS^C96Y^ and the cells expressing hINS^LB15YB16delinsH^ over the background levels (insets in Figure 1G”,H”). However, we failed to identify the extent of the differences in the signal intensity elicited by these two mutant polypeptides. These results indicated that the ectopic expression of the human mutant preproinsulins induced ER stress; thereby, activating the UPR that induces ER chaperons in *Drosophila* tissues.

### 2.2. ER-Stress Induced the Expression of Gadd45 via UPR Pathway Mediated by PERK

We then generated a transgenic line harboring *Gadd45-GFP* reporter in which the coding frame of GFP possessed a nuclear localization sequence (NLS) under a transcriptional regulatory region of the *Gadd45* gene (see Materials and Methods) to visualize the expression of *Gadd45*, a *Drosophila* orthologue of a known stress-responsible genes in mammals. Next, to confirm previous evidence that the *Gadd45* gene expression is induced via p53 in a response to DNA damage in mammalian cells [29], we induced expression of *p53* in the eye imaginal discs under the control of glass multiple reporter (GMR) promoter (*GMR-p53*, *Gadd45-GFP*). We investigated whether the GFP reporter could successfully monitor the expression of *Gadd45*. Our investigations revealed that p53 could drive the transcription of *Gadd45*, resulting in GFP fluorescence in cell nuclei of eye imaginal discs, but none in the control (*Gadd45-GFP*) discs (Figure 2A,A” and an arrow in Figure 2B,B”). These results demonstrated how the *Gadd45-GFP* reporter helped to monitor *Gadd45* transcription under stress responses.

Thus, we next examined whether the ER stress-induced expression of the GFP reporter was observed. As previously described, the ectopic expression of a dominant-negative mutant of *Drosophila* ER chaperone (Hsc70-3^DN^) is a known ER stress inducer [27]. We observed a strong GFP fluorescence in the wing pouch region expressing Hsc70-3^DN^ in wing discs using *Bx*-Gal4 driver (*Gadd45-GFP*, *Bx* > *Hsc70-3^DN^*), indicating that *Gadd45* expression was induced. By contrast, no GFP signals in the control wing discs (*Bx-Gal4/+*; *Gadd45-GFP/+*) (Figure 2C,D,H). These results suggested that *Gadd45* is an UPR target gene whose expression is also induced in response to ER stress, other than DNA damage.

It is known that ER stress activates three independent UPR signaling pathways mediated by IRE, PERK, and ATF6 (see Introduction). Hence, we induced simultaneous expression of dsRNA against either *IRE1*, *PERK*, or *ATF6* mRNA, encoding principal ER stress sensors, in wing discs expressing Hsc70-3^DN^ to determine which of these mediators are essential for *Gadd45* induction under ER stress. We observed a distinctive GFP fluorescence representing *Gadd45-GFP* expression in the disc cells, whereas no GFP signals higher than the background were observed in the control wing imaginal discs (*Bx-Gal4/+*; *Gadd45-GFP/+*) (Figure 2C,C”,D,D”,H). Further, neither the dsRNA-derived depletion of *IRE1* nor that of *ATF6* suppressed the GFP signals in the wing discs expressing Hsc70-3^DN^ (*Gadd45-GFP*, *Bx* > *Hsc70-3^DN^*, *IRE1RNAi*, or *Gadd45-GFP*, *Bx* > *Hsc70-3^DN^*, *ATF6RNAi*, respectively) (Figure 2E,G,H). In contrast, GFP fluorescence indicating *Gadd45* gene expression was significantly suppressed by dsRNA-derived depletion of PERK (*Gadd45-GFP*, *Bx* > *Hsc70-3^DN^*, *PERKRNAi*) (Figure 2F,H, *p* < 0.001, Student’s *t*-test). These observations suggested that ER stress induced *Gadd45* transcription via the PERK pathway in *Drosophila* imaginal discs. Thus, it can be further concluded that *Gadd45-GFP* is a useful reporter to monitor cellular response to ER stress.

### 2.3. Human Mutant Preproinsulin-Induced ER Stress Also Induced Gadd45 Expression

To assess whether the human mutant preproinsulins induce the UPR in a response to the ER stress, we next investigated whether the ectopic expression of hINS^C96Y^ and hINS^LB15YB16delinsH^ induced transcription of the *Gadd45* gene using the GFP reporter. No GFP signal was observed in the control wing imaginal discs (*Gadd45-GFP*, *Bx-Gal4/+*) (Figure 3A,A”). Similarly, the expression of hINS^WT^ failed to induce GFP fluorescence, indicating induction of *Gadd45* transcription in the wing pouch region of the imaginal discs (Figure 3B,B”). In contrast, the GFP signal was detected in the pouch region expressing hINS^C96Y^ and hINS^LB15YB16delinsH^ (*Gadd45-GFP*, *Bx* > *hINS^C96Y^* and *Gadd45-GFP*, *Bx* > *hINS^L39Y40delinH^*, respectively) (Figure 3C,C”,D,D”). The GFP signal in the discs expressing hINS^C96Y^ was more intense than that in hINS^LB15YB16delinsH^. These results indicate that ER stress associated with the expression of human mutant preproinsulin induced *Gadd45* transcription.

### 2.4. Induction of ER Stress by a Constitutive Expression of Human Mutant Preproinsulins in IPCs Resulted in Growth Inhibition of Drosophila

Previous studies using mammalian type 1 diabetes models that chronic ER stress in insulin-producing cells resulted in the onset and pathogenesis of diabetes [30]. Similarly, our group had previously reported that the accumulation of ER stress by the continuous expression of Hsc70-3^DN^ in IPCs resulted in the diabetes-like phenotypes such as growth inhibition, decreased mRNA level of *Dilps,* and higher glucose level in larval hemolymph [27]. Thus, we were interested in finding if a similar diabetes-like phenotype could be observed upon the ectopic expression of human mutant preproinsulins in IPCs. As mentioned earlier, the expression of human mutant preproinsulins in IPCs activates the UPR pathway. First, we observed the adult phenotypes in which hINS^WT^ was continuously expressed in IPCs throughout their development (*ilp2* > *hINS^WT^*), which resulted in significantly larger whole bodies in these flies compared with the controls (*ilp2-Gal4/+*). The average body length of male flies raised under this condition was 5.9% higher on an average than that of the controls (*n* = 94, Figure 4A). Similarly, the average area of whole wings of *ilp2* > *hINS^WT^* adult males increased by 2.0% compared with that of control (*n* = 175) (Figure 4B). These results demonstrated how the constitutive expression of the human wild-type preproinsulin in IPCs throughout development partially stimulated *Drosophila* growth.

Next, we assessed the effect of human mutant preproinsulin-overexpression in IPCs on the overall *Drosophila* growth. The average body length of the male flies with continuous hINS^LB15YB16delinsH^ expression in IPCs significantly decreased by 2.2% compared with that of the controls (*n* = 69). In contrast, the differences in body lengths between the hINS^C96Y^ group and the controls were not significant (*n* = 79) (*p* > 0.05, Student’s *t*-test, Figure 4A). Furthermore, the whole wing size of *ilp2* > *hINS^C96Y^* adult males decreased by 4.4%, compared with the controls (*n* = 125). However, the differences between the wing of hINS^LB15YB16delinsH^ males and the controls were not statistically significant (*n* = 113) (*p* > 0.05, Student’s *t*-test, Figure 4B). These data collectively suggested that the expression of the human mutant preproinsulin in IPCs resulted in partial growth inhibition of the organisms.

### 2.5. The Ectopic Expression of Human Mutant Preproinsulins Failed to Induce Apoptosis in IPCs

It has been speculated that dysfunction and/or a loss of insulin-producing cells eventually leads to human type 1 diabetes [2]. Hence, we first induced GFP with a nuclear localization sequence (NLS) exclusively in the larval IPCs to count the cell numbers to examine whether the expression of human mutant preproinsulin leads to the accumulation of ER stress, resulting in the cell death of IPCs. It was found that none of the types of human preproinsulin in IPCs reduced the number of these cells, suggesting that apoptosis was not induced (Figure 5A).

To test whether pro-apoptotic caspases are activated in IPCs expressing the human mutant polypeptides, whole-mount staining of the larval brain was carried out with an anti-cleaved caspase-3 (CC3) antibody, which recognizes active caspase-3. Consistently, we observed no anti-CC3 immunofluorescence signal over the background levels in IPCs expressing either human wild-type or the two mutant types of preproinsulin (Figure 5B–E,B”–E”). These results collectively led us to conclude that ER stress accumulation induced by targeted expression of the human mutant preproinsulins can activate UPR, but not the initiator caspase. These results could explain why the expression of the mutant peptides did not lead to cell death in the IPC.

### 2.6. The Ectopic Expression of the Human Mutant Preproinsulins Activated the JNK Pathway Less Efficiently Than That of the Stronger ER Stress Inducer, Hsc70-3^DN^

The JNK pathway is a widely known key mediator for the transduction of stress-related signals, and thereby for apoptosis induction. We had previously reported that the ER stress by Hsc70-3^DN^ expression induced apoptosis dependent on this pathway [27]. To further understand the mechanisms by which the ER stress led to apoptosis induction, we examined whether expression of the human mutant polypeptides activated the JNK pathway. First, we performed immunostaining of wing discs expressing Hsc70-3^DN^ with an antibody recognizing a phosphorylated JNK (pJNK). We observed a weak striped pattern signal of pJNK, which corresponded to a known developmental signal in the control wing discs (*Bx-Gal4/+*) (arrows in Figure 6A,A”,E) [31]. By contrast, a much more intense pJNK signal was observed in wing discs expressing Hsc70-3^DN^ (*Bx* > *Hsc70-3^DN^*) (arrow in Figure 6B”,E). Moreover, we examined whether the pJNK immunostaining signal was observed in the wing discs expressing normal or the mutant types of human preproinsulin; in those expressing the normal human peptide (*Bx* > *hINS^WT^*), an enhanced signal over the basal levels was not observed (data not shown). In contrast, we found relatively weak pJNK signal in the wing pouch region expressing hINS^C96Y^ (*Bx* > *hINS^C96Y^*). The pJNK signal was less robust than the signal in the Hsc70-3^DN^-expressing region, but it was significantly higher than that in the discs of the control flies (arrows in Figure 6A”,C”,E). Similarly, the signal in the wing discs expressing hINS^LB15YB16delinsH^ (*Bx* > *hINS^LB15YB16delinsH^*) was faint, but slightly higher than that in the control discs (arrows in Figure 6A”,D”,E). These results suggested that the ER stress induced by two human mutant preproinsulins could result in the weak activation of the JNK pathway, but the level was not strong enough to induce apoptosis in *Drosophila* tissues.

### 2.7. ER Stress Accumulated in the Wing Disc Cells Resulted in the Activation of the JAK/STAT Pathway

It is hypothesized that ER stress drives the pathogenesis of human diseases due to inflammatory signaling, thus facilitating cell death [32]. In this study, we examined whether ER stress accumulation activated the JAK-STAT pathway in *Drosophila* tissues. We used a 10XSTAT92E-GFP reporter, in which the *GFP* gene is under tandem repeats of STAT92E-binding sequences to visualize this activation of the signaling pathway [33]. Thus, the activation of the JAK-STAT pathway would lead to GFP fluorescence in our system. We then induced targeted expression of Hsc70-3^DN^, human normal or mutant type preproinsulin in the wing pouch region of the discs. As a result, a relatively intense GFP fluorescence was observed in control wing discs (*Bx-Gal4/+*; *10XSTAT92E-GFP*) (arrowhead in Figure 7A”), but not in the wing pouch region (arrow in Figure 7A”). These data support the known observation that this signaling pathway is indispensable for the normal development in this hinge region of normal wing discs [34]. Moreover, we found a more intense GFP signal in Hsc70-3^DN^-expressing regions of all the wing discs examined (*10XSTAT92E-GFP*, *Bx* > *Hsc70-3^DN^*) (*n* = 29) (arrow in Figure 7B”, but not in Figure 7A”), which suggested that the JAK-STAT pathway was activated in response to ER stress. Similarly, the ectopic expression of hINS^C96Y^ (*10XSTAT92E-GFP*, *Bx* > *hINS^C96Y^*) in the wing pouch region resulted in the induction of GFP fluorescence less frequently (33.3% of wing discs examined) (*n* = 33) (Figure 7C). In contrast, the expression of hINS^LB15YB16delinsH^ in the wing imaginal discs failed to induce GFP fluorescence over the background level (Figure 7D). These results indicated that a relatively stronger ER stress induced by hINS^C96Y^ triggered the activation of the JAK-STAT signaling pathway, which could induce multiple stress responses.

## 3. Discussion

### 3.1. Expression of Human Mutant Preproinsulin in IPCs Gives Rise to the Drosophila Growth Inhibition Reminiscent of Undernutrition in Diabetes

It has been argued that ER stress is related to the loss and/or dysfunction of pancreatic β-cells in diabetes [10,12,35]. To verify this hypothesis, Katsube and colleagues induced ER stress in IPCs of *Drosophila* larvae by ectopically expressing the dominant negative mutant ER chaperone, Hsc70-3^DN^. They observed a loss of the IPCs, accompanied by reduced Dilps production and higher glucose level in the larval hemolymph, which eventually led to growth inhibition reminiscent of undernutrition in diabetes [27]. In this study, we found that the ectopic expression of the human mutant preproinsulin, hINS^C96Y^, and hINS^LB15YB16delinsH^ resulted in a weaker but similar growth inhibition phenotype as observed in the adults having the Hsc70-3^DN^– expressing IPCs. Either of the mutant human insulins failed to be secreted from cultured pancreatic β cells [3] or from *Drosophila* IPCs (this study). Therefore, we interpret that the mutant polypeptides were accumulated in ER of the IPC, and thereby induced ER stress. Nevertheless, the caspase activation or the loss of IPCs was not observed. Thus, we speculate that the dysfunction of IPCs, rather than affecting their viability is involved in this growth inhibition. Ectopic expression of human mutant preproinsulin induced a less severe phenotype than Hsc70-3^DN^ [27]. In contrast to the apoptosis induced by direct inhibition of the ER chaperones, the accumulation of misfolded polypeptides is more similar to the physiological condition that occurs in pancreatic β-cells in diabetes patients.

In mammals, C/EBP-homologous protein (CHOP), an essential target of the PERK-ATF4 pathway, is implicated in mediating ER stress-triggered apoptosis [36]. However, no apparent homologs of *CHOP* genes are conserved in the *Drosophila* genome [37]. Instead, ER stress caused by mutant Rhodopsin (*Rh1^G69D^*) triggered apoptosis via a signaling cascade mediated by CDK5, MEKK1, and JNK [37,38]). Among these, JNK is a well-known key mediator for the transduction of a stress signal and induction of apoptosis in response to activation of several types of stress [39]. Katsube and colleagues reported that the ectopic expression of Hsc70-3^DN^ induced apoptosis via activation of the JNK pathway [27]. Furthermore, here we demonstrated that a relatively stronger ER stress by hINS^C96Y^ expression triggered activation of the JNK signaling pathway. By contrast, a lesser ER stress by hINS^LB15YB16delinsH^ failed to do so. Although a relationship between the growth inhibition due to hINS^C96Y^ expression in IPCs and the subsequent JNK activation is uncertain, it is possible to consider that this activation can occur because of various cell stress stimuli, including protein synthesis inhibition in human IPCs [40]. Hence, it would also be interesting to investigate whether expression of endogenous *Drosophila* insulin-like peptides (Dilps) is inhibited in the IPCs expressing hINS^C96Y^.

### 3.2. Gadd45 Transcription Is Induced in Response to ER Stress in Drosophila Tissues

*Gadd45* (*Growth Arrest and DNA Damage-inducible 45*) is a stress-responsive gene that plays a crucial role in DNA repair, cell cycle arrest, and apoptosis in mammalian cells [41,42]. Recently, a line of studies suggested a correlation between Gadd45 expression and activation of UPR caused by ER stress. In a previous study using the human lung cancer cell line PC-9, it was shown that CHOP which mediates ER stress-induced apoptosis promotes the expression of Gadd45 [43]. In addition, one of the Gadd45 family proteins, Gadd45α, is induced by ER stress through the PERK-eIF2α pathway in mouse embryonic fibroblasts (MEFs) [16]. Here, we demonstrated that these findings in cultured cells were able to apply to organisms using *Drosophila*. Several studies have linked Gadd45 family proteins with a pro-apoptotic function [41,44]. In contrast, there is another evidence suggesting that Gadd45 plays a role in cell survival. Gadd45α and Gadd45β cooperate to keep hematopoietic cells alive longer after UV irradiation [45]. Although Gadd45 was shown to be one of the ER stress-inducible genes, and to plays a vital role in stress signaling, its function in ER stress response remains uninvestigated. Thus, it is necessary to examine whether the induction of Gadd45 expression results in apoptosis or cell survival in *Drosophila* tissues.

It was also reported that Gadd45β overexpression inhibited JNK activation and suppressed IL-1β-induced apoptosis in the insulinoma cells [46,47]. Thus, these findings suggested that Gadd45 may have a cell survival function in insulin-producing cells under diabetic conditions. Consistently, in a previous study using *Drosophila*, the flies with Gadd45 overexpression in the nervous system showed a higher level of stress resistance than flies without overexpression [15,48,49]. As *Drosophila* IPCs are neurosecretory cells, it is possible to consider that Gadd45 prevents these cell types from accumulating ER stress and minimizes the deleterious effect of the stress. This possibility leads to the scope of further investigations in future studies.

### 3.3. Gadd45-GFP Is a Useful Reporter for the Detection of ER Stress Accumulation and UPR Induction

A previous study had reported that Gadd45α expression is induced by ER stress through the PERK-eIF2α pathway in mammalian culture cells [16]. Consistent with this result, our experiments using Gadd45-GFP reporter demonstrated that ER stress led to the induction of *Gadd45* transcription via the PERK-mediated branch of UPR in *Drosophila* tissues. These results also indicate that the Gadd45-GFP reporter is useful as a novel reporter that detects ER stress. Besides, previous studies have reported that reactive oxygen species (ROS) trigger the activation of the PERK-eIF2α-CHOP pathway in mammalian cells [50,51]. Similarly, we have shown a significant accumulation of ROS in *Drosophila* tissues under ER stress conditions [52]. ER stress stimulates p53 expression through NF-κB activation [53]. Thus, *Gadd45* transcription is precisely regulated by p53 in mammalian cells. Consistently, we demonstrated using the *Gadd45-GFP* reporter that *Gadd45* transcription is also induced by p53 in *Drosophila* tissues. Taken together our current findings with previous results collectively, Gadd45 could be induced potentially as a consequence of oxidative stress response and inflammatory response possibly mediated by p53, associated with ER stress. Therefore, it is important to examine whether Gadd45-GFP expression is affected in tissues having ER stress accumulation in *p53* mutant background to verify this possibility.

### 3.4. Accumulation of ER Stress by hINS^C96Y^ Expression Triggers Activation of JNK and JAK-STAT Signaling Pathways

Recent studies argued that ER stress-induced UPR signaling also participates in the induction of inflammation, which exacerbates conditions of several diseases such as diabetes, obesity, atherosclerosis, and cancer [54]. In this study, we found that the JAK-STAT pathway, which is also related to an inflammatory response [55], was activated in *Drosophila* tissues having ER stress induced by either Hsc70-3^DN^ or hINS^C96Y^ expression. This result corroborates the previous findings that JAK1-STAT3 signaling is activated in mouse astrocytes having ER stress is accumulation. This phenomenon is reported to be dependent on PERK-mediated UPR signaling [56]. However, it is uncertain whether UPR can directly induce activation of the JAK-STAT pathway. Alternatively, the activation of the JAK-STAT pathway occurs to regenerate tissues lost by apoptosis. During the regeneration of larval imaginal discs mechanically wounded at larval to pupal stages, JAK-STAT signaling is also activated via a stress-responsible kinase, JNK [57,58,59]. Consistently, the ectopic expression of Hsc70-3^DN^ can induce apoptosis via strong activation of JNK pathway and pro-apoptotic caspases [27]. Therefore, to verify these two possibilities, it is important to examine whether activation of the JAK-STAT pathway occurs dependent on JNK signaling, by depletion of the JNK pathway components in tissues having ER stress accumulation.

## 4. Materials and Methods

### 4.1. Drosophila Stocks and Husbandry

Flies were maintained on normal fly food at 25 °C. Normal fly food is composed of 7.2 g agar, 40 g cornmeal, 40 g dry yeast, and 100 g glucose in 1 L water, as described previously [60]. Experiments were performed using third instar larvae raised on 28 ºC for Gal4 depending-expression. *P{GMR-p53.Ex}3* (*GMR-p53*) (#8417) for induced expression of wild type p53 in the eye imaginal discs [61] was distributed from Bloomington *Drosophila* Stock Center (BDSC, Indiana University, Bloomington, IN, USA). The following Gal4 lines were observed from BDSC and used for the ectopic gene expression: *P{ilp2-GAL4.R}2 (ilp2-Gal4)* (#37516) for IPC-specific expression, and *P{GawB}Bx^MS1096-KE^*(*Bx-Gal4*) (#8860) for the pouch region-specific expression in wing discs. *w^1118^* from BDSC was used as a normal control stock. *Bx-Gal4* and *ilp2-Gal4* lines were crossed with *w^1118^* flies to generate control larvae or adults (*Bx-Gal4/+*, *ilp2-Gal4/+*, respectively).

The following UAS lines were distributed from BDSC: *P{UAS-GFPnls}8* (*UAS-GFPnls*) (#4776), and *P{UAS-Hsc70-3.D231S}D* (*UAS-Hsc70-3^DN^*) (#5841) [27]. *P{USA-Xbp1-EGFP. LG}* (*UAS-Xbp1-GFP*) was a gift from Dr. H. Steller (Rockefeller University) [37]. The following UAS-RNAi lines were distributed from BDSC: *UAS-PERKRNAi [TRiP.GL00030]* (#35162) [62], *UAS-ATF6RNAi [TRiP.JF02109]* (#26211), Another UAS-RNAi line, *P{GD3071}* (*UAS-IRE1RNAi*) was obtained from Vienna *Drosophila* Resource Center (VDRC, Vienna, Austria). *P{10XSTAT92E-GFP}1* (*10xSTAT92E-GFP*) (BDSC, #26197) was used for monitoring the activation of *Drosophila* JAK/STAT pathway [33].

For induced expression of either human wild-type (hINS^WT^) or mutant preproinsulins (hINS^C96Y^ and hINS^L39Y40delindH^) in *Drosophila* tissues, we established transgenic lines harboring cDNA for the preproinsulins under UAS. These human preproinsulin cDNAs [3] were kindly provided by Prof. F. Barbetti (Rome Tor Vergata University). We amplified each cDNA fragment by PCR using the following set of primers:

human insulinFW 5′-CTTGAATTCGTCCTTCTGCCATGGCCCTG-3′,

human insulinRV 5′-GCATCTAGATCCATCTCTCTCGGTGCAGG-3′.

After the PCR fragment was digested with EcoRI and XbaI, the purified cDNA fragment was inserted into *pUASt* vector digested by same enzymes [63]. For a germline transformation, the resultant plasmid DNA was provided for micro-injection into *w^1118^* early embryos before cellular blastoderm stage in BestGene Co. (Chino hills, CA, USA)

For monitoring the Gadd45 expression, we generated *Gadd45-GFP* line. We amplified genomic DNA fragment containing 2 to 2631 bp upstream of a tentative transcription initiation site (corresponding to 5′ end of the longest cDNA clone) by PCR using the following set of primers:

Gadd45FW 5′-TGAGGTACCGACTCATTTGG-3′,

Gadd45RV 5′-TTGCTCGAGTTATCCGTTTG-3′.

After the PCR fragment was digested with XhoI and KpnI, the purified DNA fragment encompassing tentative upstream sequences of *Gadd45* gene was inserted into *pH-stinger* vector harboring cDNA encoding EGFPnls after the multiple cloning sites, digested by XhoI and KpnI [64]. For a germline transformation, the resultant plasmid DNA was provided for micro-injection into *w^1118^* early embryos before cellular blastoderm stage in BestGene Co. (Chino Hills, CA, USA). To monitor the Gadd45 expression in wing imaginal discs, we prepared the disc samples as described below.

### 4.2. GFP Reporter Assay

To detect activation of *Drosophila* JAK/STAT pathway in wing imaginal discs, we observed expression of GFP regulated by tandem repeats of 441bp genomic region from the first intron of *Socs36E* gene, which contains two potential STAT92E binding sites [33]. Wing discs were collected from mature third instar larvae in 1xPBS. The discs were fixed in 3.7% formaldehyde for 15 min and washed in 1xPBS thoroughly. To detect GFP fluorescence, samples were observed under a fluorescent microscopy (Olympus, Tokyo, Japan, model: IX81). To detect the ER stress-induced UPR, we used the *UAS-Xbp1-GFP* stock, in which cDNA encoding Xbp1 pre-mRNA encoding a truncated Xbp1 precursor having an extra exon harboring a stop codon were placed under the UAS sequences [37].

### 4.3. Immunostaining Procedures

Immunostaining of imaginal discs and larval brains were carried out according to the method previously described in [27]. In this study, the following antibodies were used as primary antibodies: anti-GRP78 (StressMarq Biosciences, Cadboro Bay, Victoria, BC, Canada), rabbit anti-Cleaved Caspase-3 (Asp175) (9661, Cell Signaling, Danvers, MA, USA), and anti-phospho-JNK (pThr183, pTyr185) (Calbiochem Co., La Jolla, CA, USA). The samples were observed under a fluorescent microscope model IX81 (Olympus Co., Tokyo, Japan). Image acquisition was performed using the Metamorph software version 7.6 (Molecular Devices, San Jose, CA, USA), and processed with ImageJ or Adobe Photoshop CS3 (Adobe Japan, Tokyo, Japan). Quantification of immunofluorescence intensities in the wing discs, in which Bx-Gal4-dependent gene expression occurred, were performed using image-J software (NIH Image, Bethesda, MD, USA). The intensity values calculated were normalized as 1.0 [27]. Comparisons of the two groups were carried out using the Student’s *t*-test. A *p* value of less than 0.05 was considered to be significant.

### 4.4. Counting of IPCs

IPC nuclei were labelled by IPC-specific expression of GFP carrying NLS and counted the number of GFP-positive nuclei in whole brains as described previously [27]. Larval brains of third instar larvae were prepared and fixed in formaldehyde, and subsequently mounted in Vectashield (Vector Laboratories, Burlingame, CA, USA). The samples were observed with a fluorescent microscope IX81 (Olympus, Tokyo, Japan). More than 20 larvae of each genotype were examined for IPC counting. Bar graphs were created using GraphPad Prism 6 (GraphPad Software, San Diego, CA, USA). Each dataset was assessed by Mann-Whitney’s *U*-test, as previously described [27]. When the *p*-value was less than 0.05, it was calculated using Mann–Whitney’s *U*-test of unequal variance.

### 4.5. Measurement of Adult Body Length and Wing Area

Images of adult whole bodies and gross area of wings were captured using a Nikon DIGITAL SIGHT camera (Nikon Co., Tokyo, Japan). The adult body length from the anterior end of the head to the posterior end of the abdomen was measured using the manual measurement system of Nikon DIGITAL SIGHT, according to Katsube et al. [27]. Wing area of each image was measured using Image-J software (NIH Image, Bethesda, MD, USA). The two groups were statistically compared with each other using the Student’s *t*-test. Data were considered significant at *p*-value < 0.05.

## 5. Conclusions

To understand the mechanism how the ER stress by abnormal preproinsulins that failed to be secreted from β-cells leads to onset of diabetes, we induced expression of human insulin genes carrying mutations responsible for PNDM in *Drosophila.* Normal human preproinsulin stimulated growth of whole fly bodies and tissues, when it was induced in the IPCs. By contrast, the mutant peptides which failed to be secreted from the β-cells did not stimulate the growth. Instead, they induced expression of known UPR markers, while neither of them induced apoptosis in the tissues. We newly found that the ER stress induced the expression of *Gadd45*, known as a DNA-damage responsible gene, and activated the JAK-STAT signaling. Collectively, we speculate that the growth inhibition by the mutant preproinsulins resulted from the dysfunction of IPCs, rather than apoptosis.

## Figures and Tables

**Figure 1 ijms-22-12038-f001:**
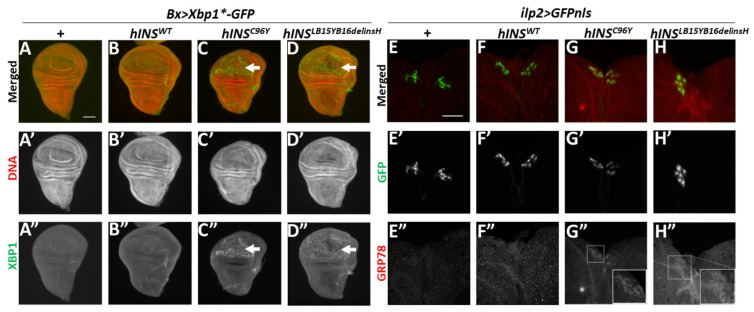
Induction of UPR observed in wing imaginal discs and larval IPCs harboring the ectopic expressing human mutant preproinsulin. (**A**–**D**) Xbp1-GFP fluorescence (green in (**A**–**D**), white in (**A”**–**D”**), arrows in (**C**,**C”**) and (**D**,**D”**)) produced because of ER stress-dependent splicing of *Xbp1*-GFP* mRNA in wing imaginal discs. Red in (**A**–**D**) (white in (**A’**–**D’**)): DNA staining. (**A**–**A”**) A control wing imaginal disc (*Bx-Gal4/+*; *UAS-Xbp1*-GFP*). (**B**–**B”**). A wing imaginal disc expressing hINS^WT^ in the wing pouch region (*Bx* > *hINS^WT^*, *Xbp1*-GFP*). (**C**–**C”**) A wing imaginal disc expressing hINS^C96Y^ (*Bx* > *hINS^C96Y^*, *Xbp1*-GFP*). (**D**–**D”**) A wing imaginal disc expressing hINS^LB15YB16delinsH^ (*Bx* > *hINS^LB15YB16delinsH^*, *Xbp1*-GFP*). Scale bar: 100 μm. (**E**–**H**) Anti-GRP78 immunostaining (red in (**E**–**H**), white in (**E”**–**H”**)) of brains the ectopically expressing GFPnls in IPCs from third-instar larvae. IPC nuclei are visualized by GFP expression (green in (**E**–**H**), white in (**E’**–**H’**)). (**E**–**E”**) A control larval brain (*ilp2* > *GFPnls*). (**F**–**F”**) A Larval brain with IPC-specific expression of hINS^WT^ (*ilp2* > *hINS^WT^*, *GFPnls*). (**G**–**G”**) A Larval brain expressing hINS^C96Y^ in IPCs (*ilp2* > *hINS^C96Y^*, *GFPnls*). (**H**–**H”**) A Larval brain expressing hINS^LB15YB16delinsH^ in IPCs (*ilp2* > *hINS^LB15YB16delinsH^*, *GFPnls*). Insets in (**G”**) and (**H”**) present a magnified view of the IPC cluster. Scale bar: 50 μm.

**Figure 2 ijms-22-12038-f002:**
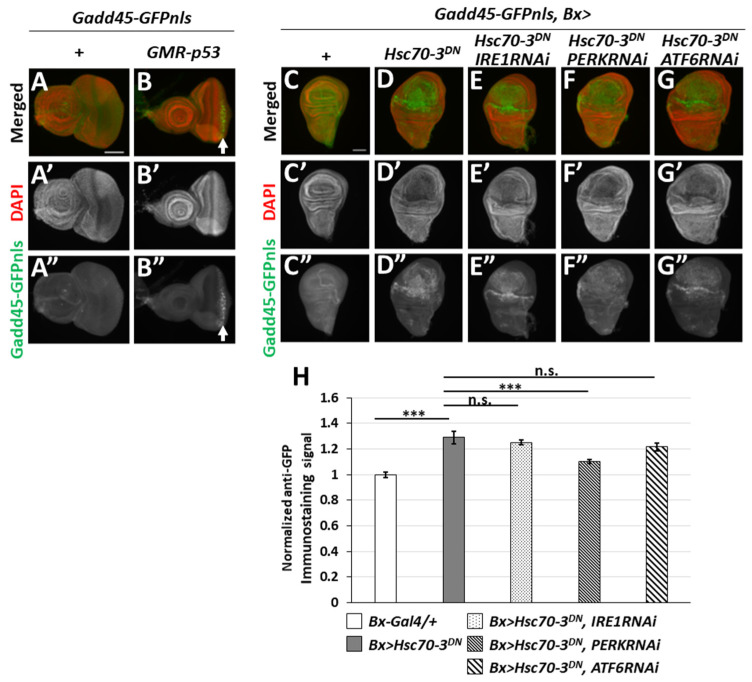
p53-Dependent Gadd45 expression and a requirement of three ER stress sensors for ER-stress induced Gadd45 expression in wing imaginal discs. (**A**,**B**) Induction of Gadd45-GFP (green in (**A**,**B**), white in (**A”**,**B”**), arrow in (**B**,**B”**)) generated by p53-dependent cellular stress response. Red in (**A**,**B**) (white in (**A’**,**B’**)): DNA staining. (**A**–**A”**) A control eye imaginal disc (*Gadd45-GFP*). (**B**–**B”**) An eye imaginal disc expressing p53 in after a morphogenic furrow of eye imaginal discs under the GMR promoter. Scale bar: 100 μm (**C**–**G**). Fluorescent micrographs of wing imaginal discs harboring Gadd45-GFP reporter. DAPI staining (red). (**C**–**C”**) A Control wing disc (*Bx-Gal4/+*; *Gadd45-GFP/+*). (**D**–**D”**) A wing disc having Hsc70-3^DN^ expression in the wing pouch region (*Gadd45-GFP*, *Bx* > *Hsc70-3^DN^*). (**E**–**E”**) A wing disc having Hsc70-3^DN^ expression and IRE1 depletion (*Gadd45-GFP*, *Bx* > *Hsc70-3^DN^*, *IRE1RNAi*). (**F**–**F”**) A wing disc having Hsc70-3^DN^ expression and PERK depletion (*Gadd45-GFP*, *Bx* > *Hsc70-3^DN^*, *PERKRNAi*). (**G**–**G”**) A wing disc having Hsc70-3^DN^ expression, ATF6 depletion (*Gadd45-GFP*, *Bx* > *Hsc70-3^DN^*, *ATF6RNAi*). Scale bar: 100 μm. (**H**) Quantification of GFP levels in wing discs. The intensity of Gadd45-GFP fluorescence in each wing imaginal disc expressing Hsc70-3^DN^ and/or dsRNA against each of three ER stress sensors as calculated and normalized to that of the control, set to 1.0 (*Bx-Gal4/+*) (*n* > 21, *** *p* < 0.001, n.s.: not significant, Student’s *t*-test). Error bars represent SEMs.

**Figure 3 ijms-22-12038-f003:**
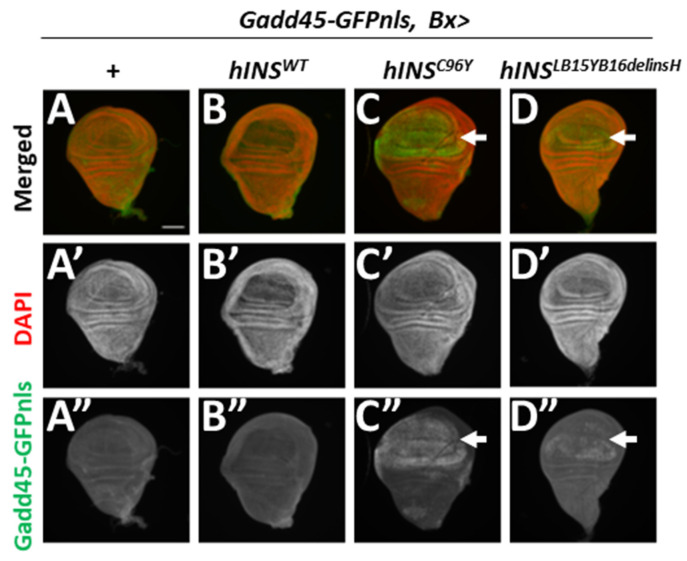
The ectopic expression of human mutant preproinsulin, hINS^C96Y^, and hINS^LB15YB16delinsH^ induced *Gadd45* transcription in wing imaginal discs. (**A**,**B**) GFP fluorescence (green in (**A**,**B**), white in (**A”**–**D”**), arrow in (**C**,**C”**) and (**D**,**D”**)) indicating Gadd45 expression in wing imaginal discs. Red in **A**–**D**: DNA staining (white in **A’**–**D’**). (**A**–**A”**) A control wing disc (*Bx-Gal4/+*; *Gadd45-GFP/+*). (**B**–**B”**) A wing disc expressing hINS^WT^ in the wing pouch region (*Gadd45-GFP*, *Bx* > *hINS^WT^*). (**C**–**C”**) A wing disc expressing hINS^C96Y^ (*Gadd45-GFP*, *Bx* > *hINS^C96Y^*). (**D**–**D”**) A wing disc expressing hINS^LB15YB16delinsH^ (*Gadd45-GFP*, *Bx* > *hINS^LB15YB16delinsH^*). Scale bar: 100 μm.

**Figure 4 ijms-22-12038-f004:**
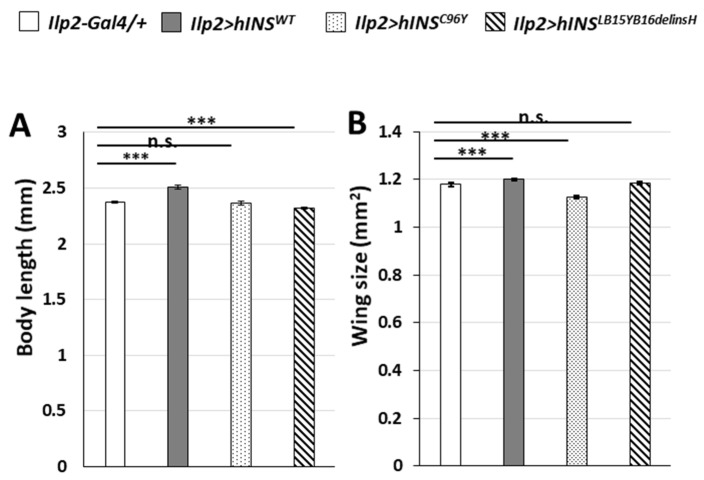
The ectopic expression of human mutant preproinsulin in IPCs resulted in the production of smaller adults indicating growth inhibition. (**A**,**B**) Adult phenotypes in male flies derived from individuals harboring targeted expression of human preproinsulin in IPCs throughout development. Note that the expression of hINS^WT^ in IPCs stimulated *Drosophila* growth, while either of hINS^C96Y^ or hINS^LB15YB16delinsH^ failed. (**A**) Quantification of body length of male adults (*n* > 69, n.s.: not significant, *** *p* < 0.001, Student’s *t*-test). Error bars represent SEMs. (**B**) Quantification of wing size of the male adults (*n* > 113, n.s.: not significant, *** *p* < 0.001, Student’s *t*-test). Error bars represent SEMs.

**Figure 5 ijms-22-12038-f005:**
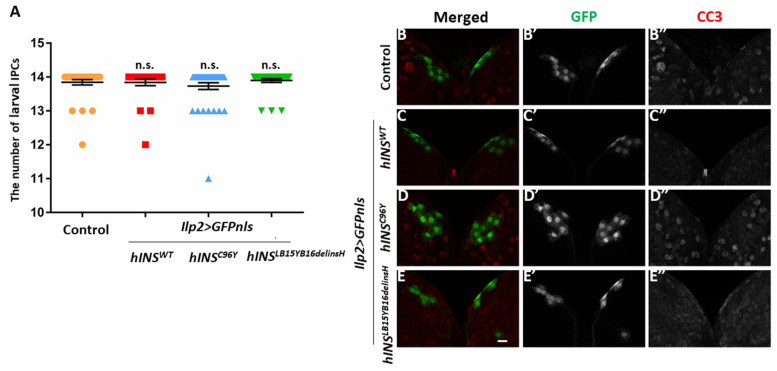
The ectopic expression of neither of hINS^C96Y^ or hINS^LB15YB16delinsH^ induced caspase activation or the loss of larval IPCs. (**A**) The number of larval IPCs in control larvae and larvae expressing human preproinsulin at the third-instar stage. Note that the differences in the IPC number between larvae expressing every three types of preproinsulin and control larvae (*ilp2* > *GFPnls*) were not significant (*n* > 25, n.s.: not significant, Mann–Whitney’s *U*-test). (**B**–**E**) Immunostaining of third-instar larval brains having IPCs labeled with GFPnls with anti-Cleaved Caspase-3 (CC3) antibody. Red in B–E (white in (**B”**–**E”**)): Anti-CC3 immunostaining signal. Green I n (**B**–**E**) (white in (**B’**–**E’**)): IPCs visualized by expression of GFPnls. (**B**–**B”**) Control larval brain (*ilp2* > *GFPnls*). (**C**–**C”**) Larval brain the ectopic expressing hINS^WT^ in IPCs (*ilp2* > *hINS^WT^*, *GFPnls*). (**D**–**D”**) Larval brain with IPCs-specific expression of hINS^C96Y^ (*ilp2* > *hINS^C96Y^*, *GFPnls*). (**E**–**E”**) Larval brain with IPCs-specific expression of hINS^LB15YB16delinsH^ (*ilp2* > *hINS^LB15YB16delinsH^*, *GFPnls*). Scale bar: 10 μm.

**Figure 6 ijms-22-12038-f006:**
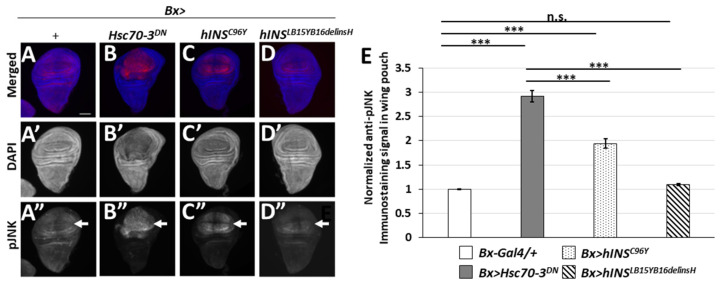
The ectopic expression of hINS^C96Y^ can activate JNK pathway in wing imaginal discs. (**A**–**D**) Anti-pJNK immunostaining of wing imaginal discs having an accumulation of ER stress in third-instar larvae. (**A**–**A”**) A control wing disc (*Bx-Gal4/+*). (**B**–**B”**) A wing discs expressing Hsc70-3^DN^ (*Bx* > *Hsc70-3^DN^*). (**C**–**C”**) A wing disc expressing hINS^C96Y^ (*Bx* > *hINS^C96Y^*). (**D**–**D”**) A wing disc expressing hINS^LB15YB16delinsH^ (*Bx* > *hINS^LB15YB16delinsH^*). Anti-pJNK immunostaining signal and DNA staining are colored in red (white in (**A”**–**D”**)) and blue (white in (**A’**–**D’**)), respectively. Scale bar: 100 μm. (**E**) Quantification of pJNK signals in wing imaginal discs. The intensity of anti-pJNK immunofluorescence was calculated and normalized to the control value, which was set as 1.0 (*Bx-Gal4/+*) (*n* > 15, *** *p* < 0.001, n.s.: not significant, Student’s *t*-test) Error bars represent SEMs.

**Figure 7 ijms-22-12038-f007:**
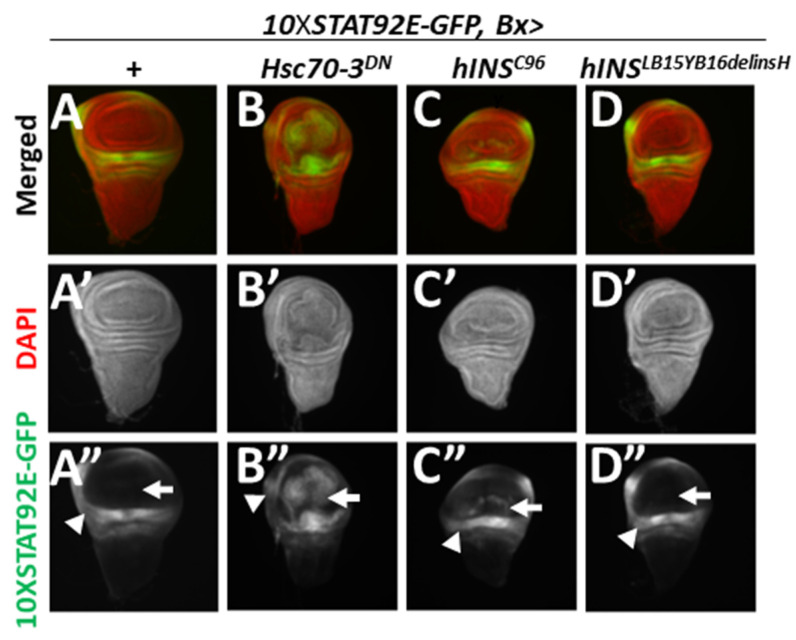
Activation of JAK/STAT signaling pathway observed in wing imaginal discs with the ectopic expression of Hsc70-3^DN^ or hINS^C96Y^. (**A**–**D**) GFP fluorescence (Green in (**A**–**D**), white in (**A**–**B”**)) indicating activation of JAK/STAT signaling pathway in wing imaginal discs. Red in (**A**–**D**) (white in (**A’**–**D’**)): DNA staining. (**A**–**A”**) A control wing imaginal disc (*Bx-Gal4/+*; *10XSTAT92E-GFP*). (**B**–**B”**) A wing disc expressing Hsc70-3^DN^ (*10XSTAT92E-GFP*, *Bx* > *Hsc70-3^DN^*). (**C**–**C”**) A wing disc expressing hINS^C96Y^ (*10XSTAT92E-GFP*, *Bx* > *hINS^C96Y^*). (**D**–**D”**) A wing disc expressing hINS^LB15YB16delinsH^ (*10XSTAT92E-GFP*, *Bx* > *hINS^LB15YB16delinsH^*).

## Data Availability

Not applicable.

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
