# Peer review of "Expression of Human Mutant Preproinsulins Induced Unfolded Protein Response, *Gadd45* Expression, JAK-STAT Activation, and Growth Inhibition in *Drosophila"

_ijms, 2021, doi:10.3390/ijms222112038_

Round 1
Reviewer 1 Report
Manuscript ijms-1416234 - (Expression of human mutant preproinsulins induced unfolded protein response, Gadd45 expression, JAK-STAT activation, and diabetes-like growth inhibition in Drosophila).
The authors expressed two types of human mutant insulins in fruit fly and studied the consequences by examining the effects on candidate gene expression. The article is well written. I have only a few minor comments for the authors:
- The authors use the term “diabetes-like growth inhibition”. To my understanding, insulin-like peptides play roles in growth and metabolism in insects, whereas these two effects are separated in mammals, in which insulin and insulin-like growth factors use different receptors. The growth inhibition observed in flies should therefore not be termed “diabetes-like”.
- The authors did not examine the secretion of recombinant mutant human insulins from Drosophila IPCs. Some of the effects that authors observe might be caused by differential binding of the mutant human insulins to insulin receptors, which could mimic or contribute to the ER stress phenotype. Another protein causing UPR in the IPCs should be used as a control to separate such effects.
- Could the authors provide the sequence alignment/differences between wt and the two mutant hINS as supplementary information?
- In line 233 please add “the wings” to the sentence: … the differences between “the wings” of hINSL39Y40delinsH males and controls…
Author Response
The authors expressed two types of human mutant insulins in fruit fly and studied the consequences by examining the effects on candidate gene expression. The article is well written. I have only a few minor comments for the authors:
- The authors use the term “diabetes-like growth inhibition”. To my understanding, insulin-like peptides play roles in growth and metabolism in insects, whereas these two effects are separated in mammals, in which insulin and insulin-like growth factors use different receptors. The growth inhibition observed in flies should therefore not be termed “diabetes-like”.
We appreciate the reviewers for their careful reading and for providing thoughtful comments. According to the reviewer’s comments, we revised the title and the sentence in line 492, page 18, so as to remove the “diabetes-like”, as requested.
- The authors did not examine the secretion of recombinant mutant human insulins from Drosophila Some of the effects that authors observe might be caused by differential binding of the mutant human insulins to insulin receptors, which could mimic or contribute to the ER stress phenotype. Another protein causing UPR in the IPCs should be used as a control to separate such effects.
After starvation, the anti-human insulin immunostaining signal of the Drosophila IPCs harbouring ectopic expression of the wild-type insulin became weaker after re-feeding the adults with the diet. This suggests that the human insulin could be released from the IPCs, according to the improved nutrition condition. By contrast, although the mutant polypeptide hINSC96Y was accumulated in the IPCs, a significant signal reduction was not observed after re-feeding the diet. This observation was consistent with the published result that the mutant insulin failed to be secreted from the human cultured cells (Colombo et al., 2008). However, as the reviewer pointed out, we have not confirmed the results directly, using the ELISA for example. In our next paper, we will present the evidences indicating that neither of the human mutant insulins, hINSC96Yor hINSLB15YB16delinsH are secreted into the hemolymph.
As the reviewer pointed out, we could neither exclude the possibility that some of the effects that we observed might be a consequence of differential binding of the mutant human insulins to insulin receptors, which could mimic or contribute to the ER stress phenotype. However, in our previous study, we showed that ectopic expression of the dominant negative mutant of the ER chaperon, which caused the UPR in the IPCs, resulted in the similar growth defect (Katsube et al., 2019). Therefore, we speculate that the mutant insulins that failed to be secreted from the b cell line would be primary defects of the ER stress phenotype.
In addition, we realized that name of the mutant insulin was described by mistake in the previous manuscript. We corrected it as hINS LB15YB16delinsH, according the original paper (Colombo et al., 2008).
- Could the authors provide the sequence alignment/differences between wt and the two mutant hINS as supplementary information?
As suggested, we added the Fig. S1 to provide the sequence alignment/differences between wt and the two mutant polypeptides, and following three sentences underlined to describe the sequence differences between wt and the two mutant hINSs as follows;
(In line 128-130, page 5) “This mutant polypeptide has a Cys to Tyr substitution at proinsulin position 96 (C96Y; encoding residue 7 of the insulin A-chain), which disrupts a disulfide bond connecting the insulin B and A-chains (Fig. S1).”
(In line 148-151, page 5-6) “In addition to the hINSC96Y described before, another human mutant insulin, hINSLB15YB16delinsH carrying an in-frame substitution, in which B-chain Leu15 and Tyr16 are replaced by a single His residue. Both mutant polypeptides failed to secrete from b-cells.”
- In line 233 please add “the wings” to the sentence: … the differences between “the wings” of hINSL39Y40delinsHmales and controls…
According to the reviewer’s comments, we added the phrase “the wings of ” in line 238 in page 9, as requested.
Reviewer 2 Report
In the current manuscript Yamazoe and colleagues use Drosophila model to show that the diabetes-like growth inhibition by the mutant preproinsulins result from the dysfunction of IPCs, rather than apoptosis. Authors first show that the expression of human preproinsulin mutants induced ER stress in IPCs. Authors then generate and validate a transgenic tool Gadd45-GFP reporter to visualize the Gadd45 expression. Authors then convincingly show that preproinsulin mutant associated ER-stress also induces Gadd45 expression and ER-stress associated Gadd45 expression activates the JAK-STAT pathway. Authors then go on to show that preproinsulin mutant associated ER-stress affects growth in flies.
I do not have any major concerns in the present work.
Author Response
We appreciate the reviewer for his/her careful reading of the manuscript.